# Arylamine Analogs of Methylene Blue: Substituent Effect on Aggregation Behavior and DNA Binding

**DOI:** 10.3390/ijms22115847

**Published:** 2021-05-29

**Authors:** Alena Khadieva, Olga Mostovaya, Pavel Padnya, Valeriy Kalinin, Denis Grishaev, Dmitrii Tumakov, Ivan Stoikov

**Affiliations:** 1A.M. Butlerov’ Chemistry Institute of Kazan Federal University, 18 Kremlyovskaya Str., 420008 Kazan, Russia; as-alex93@mail.ru (A.K.); olga.mostovaya@mail.ru (O.M.); valerargentum@gmail.com (V.K.); 2Scientific and Educational Center of Pharmaceutics, Kazan Federal University, 420008 Kazan, Russia; dionis.grishaev@yandex.ru; 3Institute of Computational Mathematics and Information Technologies, Kazan Federal University, 420008 Kazan, Russia; dtumakov@kpfu.ru

**Keywords:** methylene blue, phenothiazine, 3,7-bis(N-phenylamino)phenothiazin-5-iums, DNA, intercalation, dimerization

## Abstract

The synthesis of new phenothiazine derivatives, analogs of Methylene Blue, is of particular interest in the design of new drugs, as well as in the development of a new generation of agents for photodynamic therapy. In this study, two new derivatives of phenothiazine, i.e., 3,7-bis(4-aminophenylamino)phenothiazin-5-ium chloride dihydrochloride (**PTZ1**) and 3,7-bis(4-sulfophenylamino)phenothiazin-5-ium chloride (**PTZ2**), are synthesized for the first time and characterized by NMR, IR spectroscopy, HRMS and elemental analysis. The interaction of the obtained compounds **PTZ1** and **PTZ2** with salmon sperm DNA is investigated. It is shown by UV-Vis spectroscopy and DFT calculations that substituents in arylamine fragments play a crucial role in dimer formation and interaction with DNA. In the case of **PTZ1**, two amine groups promote H-aggregate formation and DNA interactions through groove binding and intercalation. In the case of **PTZ2**, sulfanilic acid fragments prevent any dimer formation and DNA binding due to electrostatic repulsion. DNA interaction mechanisms are studied and confirmed by UV-vis and fluorescence spectroscopy in comparison with Methylene Blue. The obtained results open significant opportunities for the development of new drugs and photodynamic agents.

## 1. Introduction

The study of drug-DNA interactions plays a key role in pharmacology, since small molecules capable of binding genomic DNA can become effective anti-cancer, antibiotic and antiviral therapeutic agents [1]. Three models of drug interactions exist with the DNA double helix [1,2,3]. First, positively charged molecule fragments can bind to negatively charged DNA phosphate groups through electrostatic interactions. This type of interaction usually occurs on the outer side of the helix. Second, drug molecules can bind to nucleic acid bases in the major and/or minor groove of the DNA helix through hydrogen bonds and van der Waals interactions. The third example of interaction with DNA is intercalation. which is π-stacking between nucleobases and molecule aromatic fragments. In this case, a flat heteroaromatic fragment is located between DNA base pairs, and binds perpendicularly to the helix axis. It is well established that intercalating fluorescent dyes allow the probing of DNA molecules. Intercalation is also the key feature of clinically used antitumor drugs. Therefore, studying the interactions of intercalative small molecules with DNA opens a significant opportunity in higher generation diagnostic probes and DNA-directed therapeutics design [4].

Methylene Blue (MB), a phenothiazine derivative, is a well-studied dye and photosensitizer used in photodynamic therapy for the treatment of cancer [5,6,7]. The mechanisms of interaction between MB and DNA have been studied in detail, which include electrostatic interaction (due to the positive charge of the molecule), minor groove binding, and intercalation [8,9,10]. Fluorescently active DNA intercalators are in demand as probes for monitoring reactions with DNA for research purposes. However, intercalation is often a limiting factor in drug development, since it may affect important processes in DNA, such as replication, transcription, and repair, which makes intercalators potent mutagens [11]. Thus, the ability to control the interaction of organic compounds with DNA is an indispensable part of modern drugs design. Another factor in the development of drugs, as well as photodynamic agents, is their aggregation ability. It is known that MB forms H-aggregates, which negatively affects its efficiency as a photodynamic agent, reducing the yield of singlet oxygen [12].

There are many reports on the synthesis of new phenothiazine derivatives, MB analogs, for the needs of modern medicine, including the design of new drugs, as well as the development of new generation photodynamic agents [13,14,15,16]. There are several reports on the synthesis of 3,7-bis(aryl-amino)phenothiazine derivatives [17,18,19,20]. Some of these report on the low cytotoxicity [20] and antibacterial activity of these compounds [19]. However, there are no data on their interaction with DNA. It turned out in a study of the effect of aromatic substituents of phenothiazine on aggregation ability that arylamine substituents at positions 3 and 7 of phenothiazine can inhibit aggregation ability. [21].

In the present work, 3,7-bis(aryl-amino)phenothiazine derivatives containing two primary amine (3,7-bis((4-aminophenyl)amino)phenothiazin-5-ium chloride dihydrochloride dihydrochloride **PTZ1**) and sulfo groups (3,7-bis((4-sulfophenyl)amino)phenothiazin-5-ium chloride **PTZ2**) are synthesized. The effect of substituents in the arylamine fragment of 3,7-bis(aryl-amino)phenothiazine derivatives on their aggregation properties and interaction with DNA is studied.

## 2. Results and Discussion

### 2.1. Synthesis of **PTZ1**, **PTZ2**

There are many reports in the literature on the synthesis of new MB derivatives, including arylamine derivatives. The introduction of aniline and its derivatives at positions 3 and 7 of phenothiazine provides a conjugated system, structurally similar to emeraldine, with unique physicochemical properties [18]. Arylamine derivatives of phenothiazine have been used in colorimetric [22], electrochemical sensors [23,24], catalysis [17], and as antibacterial agents [25]. It is possible to design supramolecular systems for directed self-assembly into binary associates by varying the substituents in the arylamine fragment of phenothiazine [26]. The structural diversity of these derivatives remains low despite their high potential. The main approaches to the introduction of arylamine fragments into the structure of phenothiazine are the Buchwald-Hartwig reaction [18,27,28] and the oxidation of phenothiazine to phenothiazin-5-ium cation followed by nucleophilic addition of aromatic amines (Scheme 1) [17,21,25,29].

For the Buchwald-Hartwig reaction, substituted at the nitrogen atom (position 10, Scheme 1), phenothiazine derivatives are used as a starting compound which prevents the formation of the cationic form. Therefore, this method does not suit us. A simple two-stage method for the preparation of 3,7-bis(aryl-amino)phenothiazine derivatives (Scheme 2) was implemented for the synthesis of target compounds **PTZ1** and **PTZ2**, namely, the oxidation of phenothiazine by molecular iodine to phenothiazin-5-ium tetraiodide (**PTZI4**) followed by nucleophilic addition of aromatic amines. The resulting compound **PTZa** was involved in hydrolysis reaction with concentrated hydrochloric acid to obtain the target compound **PTZ1**. The hydrochloric acid treatment also resulted in the replacement of the iodide anion with chloride.

The structure and composition of all obtained phenothiazine derivatives were confirmed by a series of physical methods, namely, ^1^H and ^13^C NMR spectroscopy, IR spectroscopy, high resolution mass spectrometry (HRMS), and elemental analysis (Appendix A).

The synthesis of the target compounds **PTZa**, **PTZ1**, **PTZ2** is evidenced in the ^1^H NMR spectra by the signals of the aromatic protons of the phenothiazin-5-ium fragment, namely, doublet of protons H1 and H9 with chemical shift 8.05 ppm (**PTZa**), 8.11 ppm (**PTZ2**), doublet of protons H2 and H8 with chemical shift in the region of 7.68–7.50 ppm and a singlet of protons H4 and H6 in the region of 7.68–7.50 ppm. Vicinal proton–proton coupling constants of phenothiazin-5-ium fragment are ^3^*J*_HH_ = 9.3 Hz, while for the aromatic substituents ^3^*J*_HH_ is always less than 9 Hz [17,26,30], which makes it possible to identify the signals of the protons of the phenothiazin-5-ium fragment in ^1^H NMR spectra. In the ^1^H and ^13^C NMR spectra of the target compound **PTZ1**, a **PTZa** hydrolysis product, no signals of protons and carbons of methyl fragments of acetamide groups are observed. In ^1^H NMR spectra of **PTZ1**, signals of the aromatic protons upfield shifted to 6.61–7.84 ppm due to the electron donor effect of NH_2_ groups, caused by partial deprotonation with solvent. Signals of protons of NH groups also shifted to 8.49 ppm. A broad singlet is also observed in the region of 5.3 ppm, which can be attributed to the signals of NH_3_+ groups’ protons.

In the IR spectra of **PTZa**, **PTZ1**, **PTZ2,** absorption bands corresponding to the phenothiazine-5-ium fragment are observed in regions 1568–1580 cm^−1^ (C-N), 1360–1377 cm^−1^ (C=S^+^), 1115−1125 cm^−1^ (C-N) и 675−685 cm^−1^ (C-S). For the compound **PTZa** with an acetanilide fragments, there is a band at 1160 cm^–1^, which corresponds to the stretching vibrations of C=O bonds. This band is absent in the IR spectrum of the compound **PTZ1**, which indicates complete hydrolysis of acetamide fragments. **PTZ1** is characterized by several broad bands in the region of 3000–2900 cm^−1^, corresponding to bending vibrations of the N-H bond of the NH_3_+ fragment. The **PTZ2** compound with sulfanilic acid fragments is characterized by absorption bands with maxima at 1142, 1115, 1029, 1002 cm^–1^, which corresponds to the vibrations of the sulfanilic acid fragment.

In high-resolution mass spectra (HRMS) of compounds **PTZa**, **PTZ1**, **PTZ2**, the main peaks correspond to the molecular ions of these compounds within an accuracy of 0.0005 *m/z*, which indicates a high purity of the target compounds.

### 2.2. Aggregation Properties—Dimer Formation

The aggregation of cationic dye molecules leads to changes in their photophysical properties. Aggregation of phenothiazine derivatives can be used for targeted design of materials (binary co-crystals, π-complexes) [31,32,33]. However, this can also lead to undesirable effects, such as a decrease in the yield of singlet oxygen [12]. Previously, MB aggregation processes with the formation of H and J aggregates were studied in detail [34,35]. In this regard, we first studied the aggregation properties of compounds **PTZ1** and **PTZ2** in comparison with MB (Figure 1).

It is known that MB has two aggregate forms in solution, i.e., monomer (λ_max_ = 664 nm) and dimer (λ_max_ = 613 nm) (Figure 1A) [34]. The **PTZ1** compound (10 mM Tris-HCl buffer) showed a broad absorption band with λ_max_ = 630 nm, with a low extinction coefficient which indicates self-association of the compound into H-dimers such as MB and other binary systems, based on phenothiazine derivatives described in the literature [36]. We found the conditions for obtaining monomeric forms, i.e., **PTZ1** was dissolved in a Tris-HCl buffer solution containing 1 vol.% of DMSO. The monomeric structure of the product is indicated by a red shift of the absorption band of **PTZ1** (up to 669 nm). To confirm the formation of **PTZ1** H-aggregates, quantum-mechanical calculations were performed in Orca software (BP86/def2-SVP, def2/J). The geometries of the three types of conformers were determined (Figure 2), as well as the geometry of the H-dimer. The dimer formed by π-stacking of type A conformers is characterized by the lowest energy (−3195.1675 Hartrees). The distance between the planes of the heterocycles is 3.5 Å (Figure 3). The geometry of the H-dimer of **PTZ1** is similar to that of the H-dimer of MB [21].

There is a single absorption band with λ_max_ = 669 nm in the UV-Vis spectrum of the compound **PTZ2**. In spite of varying concentrations, the molar extinction coefficient remains constant and is approximately 40,000 M^−1^ × cm^−1^. This indicates that **PTZ2** does not form associates in an aqueous solution. This effect is achieved due to the electrostatic repulsion of sulfonic acid fragments.

Thus, the effect of substituents in the aromatic fragment of phenothiazine-5-ium derivatives (**PTZ1**, **PTZ2**) on their aggregation properties was studied by UV-Visible spectroscopy. The compound **PTZ1** containing two *p*-phenylenediamine fragments forms H-dimers such as MB, while **PTZ2** does not form any associates in solution due to the electrostatic repulsion of sulfonic acid groups.

### 2.3. Interaction with DNA

It is known that many phenothiazine derivatives, including MB, are DNA intercalators [37] due to the flat geometry and positive charge of the phenothiazin-5-ium fragment. In this regard, the interaction of the synthesized derivatives of **PTZ1** and **PTZ2** with model salmon sperm DNA was studied by UV-Vis and fluorescence spectroscopy. The studies were carried out in comparison with the MB.

#### 2.3.1. UV-Vis Spectroscopy

Initially, the interaction of the compounds **PTZ1** and **PTZ2** with salmon sperm DNA was studied by UV-Vis spectroscopy in 10mM Tris-HCl buffer (pH = 7.4). To determine the type of interaction and binding constants, the compounds **PTZ1**, **PTZ2**, and MB were titrated with DNA solution. The interaction was monitored in the long-wavelength region of the spectrum (500–800 nm), where only dyes absorb (Figure 4). 

In the observed range of [MB]/[DNA] concentration ratios from 3 to 0.3 (DNA concentration in nucleotide base pairs), an increase in DNA concentration leads to a decrease in MB absorption (hypochromic effect) without a shift of absorption band maximum (Figure 4A). This process is due to the DNA minor groove binding of MB. As shown in the literature [8,38], higher DNA concentrations ([MB]/[DNA] < 0.14) lead to hypochromic effect with bathochromic shift, which indicates a strong intermolecular interaction, including effective overlap of the p-electron cloud with polynucleotide base pairs typical for “intercalation ligand-DNA” complexation.

We determined the binding constant by a graphical method (Appendix A) according to the Benesi—Hildebrand equation [39] (K_b_ = 1.67 × 10^4^ M^−1^) which is in agreement with data obtained using this method [40]. The binding constant was also determined using Bindfit (K’_b_ = 2.42 × 10^4^ M^−1^, stoichiometry 1:1) [41,42,43]. We chose two methods for determining the dye–DNA binding constants: (1) a graphical method based on the Benesi-Hildebrand equation as a traditional method for dyes of a number of phenothiazine derivatives and (2) a modern automated method using Bindfit application based on performing a non-linear regression on titration data via a python program on its server. The results obtained by the two methods correlate with each other.

Further, the interaction between salmon sperm DNA with monomeric and dimeric forms of **PTZ1** was studied. A bathochromic shift and hyperchromic effect are observed when DNA is added to a solution of dimeric forms of **PTZ1** (Figure 4C). It is interesting that the interaction of the compound **PTZ1** with DNA prevents dimer formation. Similar effects were reported in the literature [44]. The phenothiazine derivative containing a bispiperazine linker in the absence of DNA is characterized by λ_max_ = 620 nm, while in the presence of DNA or sodium dodecyl sulfate (an agent that prevents dimerization), λ_max_ = 676 nm. In the observed range of DNA concentrations, no isosbestic point is observed which indicates several processes, i.e., dimer decomposition and the interaction of monomers with DNA.

The study of the interaction of monomeric forms of **PTZ1** with DNA by spectrophotometric titration revealed two main ranges of [**PTZ1**]/[DNA] concentration ratios, indicating the interaction of the dye with DNA similar to MB [8,38] (Figure 4D): (1) only a hypochromic effect is observed at [**PTZ1**]/[DNA] from 10 to 1.75, (2) and a bathochromic shift and a hyperchromic effect are observed (hypochromic in relation to free form of the dye) at [**PTZ1**]/[DNA] from 1.75 to 0.8. The absorption spectra (1) are characterized by the presence of an isosbestic point at 742 nm, which indicates the presence of one type of binding (interaction). Therefore, the binding constant by the Benesi—Hildebrand equation (Appendix A) can be obtained, which is 1.59 × 10^4^ M^−1^. The constant was also determined using Bindfit (K’_b_ = 3.99 × 10^4^ M^−1^, stoichiometry 1:1). Both constants are close to results obtained for MB. The second process (2) is characterized by a bathochromic shift with λ_max_ = 684 nm. Based on the similarity of the processes and the values of the MB and **PTZ1** binding constants, it can be assumed that process (2) is associated with the intercalation of **PTZ1** into DNA.

With the addition of DNA to the negatively charged **PTZ2** derivative, no significant changes were observed in the absorption spectra, which indicates the absence of interaction of the compound **PTZ2** with salmon sperm DNA.

In summary, the effect of substituents in the aromatic fragment in arylamine derivatives of phenothiazine-5-ium on their interaction with DNA by UV-Vis spectroscopy was investigated. In the case of **PTZ1**, the presence of primary amino groups leads to efficient binding to DNA of dimeric and monomeric forms of **PTZ1**. The interaction of **PTZ1** dimers with DNA leads to dimer decomposition which results in bathochromic shift and hyperchromic effect in the absorption spectra. In the case of interaction of **PTZ1** monomers with DNA, hyperchromic effect with bathochromic shift was observed. The spectral properties of the interaction, as well as the binding constants of **PTZ1** with DNA, are similar to the characteristics of the interaction of MB with DNA which indicates similar mechanisms of interaction of dyes with DNA. **PTZ1**-DNA binding constant is 3.99 × 10^4^ M^−1^ (Bindfit). In the case of compound **PTZ2** containing sulfonic acid fragments in its structure, the absence of interaction of the dye with DNA was shown, which is due to the electrostatic repulsion of sulfonic acid fragments from DNA phosphate groups.

#### 2.3.2. Fluorescence Spectroscopy

One of the most convenient and reliable methods for determining the mechanism of the interaction of organic dyes with DNA is fluorescence spectroscopy. The interaction of MB with DNA, resulting in quenching of the fluorescence intensity of the dye relative to its free form, is described in detail in the literature [45].

Initially, the fluorescence spectra of compounds **PTZ1** and **PTZ2** were recorded ([**PTZ1**] = 2.37 × 10^−5^ M, [**PTZ2**] = 1.18 × 10^−5^ M). Moreover, the obtained compounds have a low fluorescence intensity. In this regard, it was not possible to determine the interaction constant by fluorescence titration. Based on the dye/DNA concentration ratios for the UV-vis spectroscopy titration for **PTZ1**, two key characteristic concentration ratios were chosen: [dye]/[DNA] > 2, where only a hypochromic effect was observed, and [dye]/[DNA] < 2, where a bathochromic shift was observed (Figure 5).

For **PTZ1** at a concentration ratio of [**PTZ1**]/[DNA] = 15, no changes in the fluorescence spectrum are observed, while for [**PTZ1**]/[DNA] = 1.5, fluorescence quenching is observed (Figure 5A). The most pronounced effect is observed when [**PTZ1**]/[DNA] < 1 (Figure 5B).

To confirm the intercalation process of compound **PTZ1** into the DNA helix, an experiment was carried out with ferrocyanide ions, which are known as “fluorescence quenchers” of free dye molecules in solution [45]. Fluorescence quenching is based on charge transfer complex formation between a dye molecule in an excited state and a ferrocyanide ion acting as an electron donor [46,47]. This experiment has already been described in the literature for MB [48]. The study of fluorescence quenching is a reliable verification of dye molecules availability for fluorescence quenchers [49,50,51]. When the dye binds in DNA groove, it remains available for the quencher in solution. In the case of intercalation, the dye located between the pairs of nucleic bases is shielded from the solvent and, therefore, is inaccessible to the quencher [52]. We chose the concentration ratio [**PTZ1**]/[DNA] = 0.2 based on the results obtained by electron spectroscopy and the proposed mechanism of intercalation in this concentration range. The addition of K_4_(Fe(CN)_6_) to a solution of free dye **PTZ1** results in significant decrease of fluorescence intensity (Figure 5D), while the emission intensity of the **PTZ1**-DNA complex, both in the presence and in the absence of a quencher, practically does not change. Thus, the mechanism of intercalation of the compound **PTZ1** into DNA helix was confirmed.

In the case of **PTZ2**, with the concentration ratios [**PTZ2**]/[DNA] = 8 and [**PTZ2**]/[DNA] = 0.84, no changes in the fluorescence intensity were observed (Figure 5C). The data obtained by UV-Vis spectroscopy are in agreement with the results of fluorescence spectroscopy. 

In summary, fluorescence spectroscopy experiments confirmed that substituents in aromatic fragments at positions 3 and 7 in phenothiazin-5-ium significantly affect the interaction of the dye with DNA. Fluorescence spectroscopy data are in agreement with the results of UV-Vis spectroscopy data. 

## 3. Materials and Methods

### 3.1. General Experimental Information

All reagents and solvents (Sigma-Aldrich, USA) were used directly as purchased or purified according to the standard procedures. The ^1^H and ^13^C-NMR spectra were recorded on a Bruker Avance 400 spectrometer (Bruker Corp., Billerica, MA, USA) (400 MHz for H-atoms) for 3–5% solutions in DMSO-*d_6_*. The residual solvent peaks were used as an internal standard. Elemental analysis was performed on the Perkin-Elmer 2400 Series II instruments (Perkin Elmer, Waltham, MA, USA). The FTIR ATR spectra were recorded on the Spectrum 400 FT-IR spectrometer (Perkin Elmer, Seer Green, Lantrisant, UK) with a Diamond KRS-5 attenuated total internal reflectance attachment (resolution 0.5 cm^−1^, accumulation of 64 scans, recording time 16 s in the wavelength range 400–4000 cm^−1^). HRMS mass spectra were obtained on a quadrupole time-of-flight (t, qTOF) AB Sciex Triple TOF 5600 mass spectrometer (AB SCIEX PTE. Ltd., Singapore) using a turbo-ion spray source (nebulizer gas nitrogen, a positive ionization polarity, needle voltage 5500 V). Recording of the spectra was performed in “TOF MS” mode with collision energy 10 eV, de-clustering potentially 100 eV and with a resolution of more than 30,000 full-width half-maximum. Samples with the analyte concentration of 5 µmol/L were prepared by dissolving the test compounds in the mixture of methanol (HPLC-UV Grade, LabScan). Melting points were determined using the Boetius Block apparatus (VEB Kombinat Nagema, Radebeul, Germany).

The UV-vis measurements were performed with a Shimadzu UV-3600 instrument (Kyoto, Japan). Quartz cuvettes with an optical path length of 10 mm were used. Absorption spectra were recorded after 1 h of incubation at 25 °C.

Fluorescence spectra were recorded on a Fluorolog 3 luminescent spectrometer (Horiba Jobin Yvon, Longjumeau, France). The excitation wavelength was selected as 650 nm. The emission scan range was 665–670 nm. Excitation and emission slits were 5 nm. Quartz cuvettes with an optical path length of 10 mm were used. Fluorescence spectra were automatically corrected by the Fluorescence program. The experiment was carried out at 298 K. Solutions of the investigated systems were measured after incubating for an hour at room temperature.

DFT calculations were performed with Orca (version number: 4.2.1) using BP86 functional and def2-SVP, def2/J basis set [53,54,55].

### 3.2. Synthesis of **PTZ1** and **PTZ2**

**PTZa**, **PTZ1** were synthesized according to reported methods with minor modifications [30]. 

#### 3.2.1. 3,7-Bis((4-acetamidophenyl)amino)phenothiazin-5-ium iodide (**PTZa**)

A solution of *p*-amino-acetanilide (0.579 g, 3.86 mmol) in methanol (20 mL) was added to a suspension of phenothiazin-5-ium tetraiodide (**PTZI4**) (0.300 g, 0.425 mmol). The mixture was vigorously stirred at room temperature for 24 h. Then 15 mL of methanol was evaporated and precipitate was formed by sedimentation with diethyl ether. The obtained precipitate was filtered off, washed with diethyl ether and dried under vacuum. 

Product yield 0.150 g (56%), m. p.: 259 °C; ^1^H-NMR (DMSO-d_6_, *δ*, ppm, *J*/Hz): 2.08 (s, 6H, CH_3_), 7.39 (d, 4H, ^3^*J*_HH_ = 8.4 Hz), 7.50–7.68 (m, 4H), 7.72 (d, 4H, ^3^*J*_HH_ = 8.3 Hz), 8.05 (d, 2H, ^3^*J*_HH_ = 9.3 Hz), 10.15–10.21 (m, 2H, NH); ^13^C-NMR (DMSO-d_6_, *δ*, ppm): 169.04, 168.87, 151.80, 139.70, 139.23, 138.92, 138.34, 137.16, 135.56, 132.64, 128.32, 127.98, 126.66, 126.32, 124.06, 123.91, 122.20, 121.43, 120.45, 120.34, 114.86, 106.95, 49.07; FTIR ATR (*ν*, cm^−1^): 2934 (acetanilide fragment), 1660 (acetanilide fragment), 1587 (C-N, phenothiazinium fragment), 1507, 1388 (phenothiazinium fragment), 1120 (C-N); Elemental analysis. The calculated values for C_28_H_24_IN_5_O_2_S were as follows: C, 54.11; H, 3.89; I, 20.42; N, 11.27; S, 5.16; found: C, 54.18; H, 3.93; I, 20.56; N, 11.34; S, 5.12; HRMS: calculated [M−I^−^]^+^: *m*/*z* = 494.1645, found [M−I^−^]^+^: *m*/*z* = 494.1650.

#### 3.2.2. 3,7-Bis((4-aminophenyl)amino)phenothiazin-5-ium chloride dihydrochloride (**PTZ1**)

The compound **PTZa** (0.610 g, 1 mmol) was dissolved in the mixture of propan-2-ol (20 mL) and concentrated hydrochloric acid (20 mL) and was refluxed for 60 h. Then the solvent was evaporated. The precipitate formed was filtered off, washed with 2 M HCl and dried under vacuum.

Product yield 0.146 g (93%), d. p.: 259 °C; ^1^H-NMR (DMSO-d_6_, *δ*, ppm, *J*/Hz): 5.36 (br. s., NH_3_^+^), 6.61–6.72 (m, 8H), 7.04–7.10 (m, 4H), 7.77 (d, 2H, ^3^*J*_HH_ = 8.0 Hz), 8.49 (s, 2H, NH); ^13^C-NMR (DMSO-d6, *δ*, ppm): 146.06, 129.09, 128.81, 127.99, 126.05, 124.30, 124.21, 123.97, 123.91, 121.09, 118.08, 116.03; FTIR ATR (*ν*, cm^−1^): 1595, 1491, 1377 (phenothiazinium fragment), 1125(C-N); Elemental analysis. The calculated values for C_24_H_22_Cl_3_N_5_S were as follows: C, 55.55; H, 4.27; Cl, 20.5; N, 13.5; S, 6.18; found: C, 55.59; H, 4.34; Cl, 20.43; N, 13.44; S, 6.2; HRMS: calculated [M−2HCl−Cl^−^]^+^: *m*/*z* = 410.1434, found [M−2HCl−Cl^−^]^+^: *m*/*z* = 410.1439.

#### 3.2.3. 3,7-Bis((4-sulfophenyl)amino)phenothiazin-5-ium chloride (**PTZ2**)

A solution of 4-aminobenzenesulfonic acid sodium salt (0.829 g, 4.25 mmol) in water was added to a suspension of phenothiazine-5-ium tetraiodide (**PTZI4**) (0.300 g, 0.425 mmol). The mixture was vigorously stirred at room temperature for 48 h. Precipitate was collected and washed with diethyl ester. Then concentrated hydrochloric acid was added to residue, and the obtained mixture was vigorously stirred at room temperature for 10 h. The precipitate formed was filtered off, washed with 2 M HCl and dried under vacuum. 

Product yield 0.150 g (60%), m. p.: >300 °C; ^1^H-NMR (DMSO-d_6_, *δ*, ppm, *J*/Hz): 7.41 (d, 4H, ^3^*J*_HH_ = 8.2 Hz), 7.57 (d, 2H, ^3^*J*_HH_ = 9.3 Hz), 7.67 (s, 2H), 7.73 (d, 4H, ^3^*J*_HH_ = 8.1 Hz), 8.11 (d, 2H, ^3^*J*_HH_ = 9.3 Hz), 11.10 (s, 2H, NH); ^13^C-NMR (DMSO-d6, *δ*, ppm): 151.83, 146.84, 139.15, 137.94, 137.50, 136.16, 127.68, 123.19, 122.43, 107.40; FTIR ATR (*ν*, cm^−1^): 1574 (C-N, phenothiazinium fragment), 1332 (C=S^+^, phenothiazinium fragment), 1261 (C-N, sulfanilic acid fragment), 1142 (SO_3_), 1115 (SO_3_), 1029 (sulfanilic acid fragment), 1002 (sulfanilic acid fragment), 810 (phenothiazinium fragment), 683 (C-S); Elemental analysis. The calculated values for C_24_H_18_ClN_3_O_6_S_3_ were as follows: C, 50.04; H, 3.15; Cl, 6.15; N, 7.29; O, 16.66; S, 16.70; found: C, 50.05; H, 3.14; Cl, 6.17; N, 7.30; O, 16.65; S, 16.69; HRMS: calculated [M−Cl^−^]^+^: *m*/*z* = 540.0352, found [M−Cl^−^]^+^: *m*/*z* = 540.0357.

### 3.3. DNA Binding Studies

The studies were conducted in 10 mM Tris-HCl (pH = 7.4) buffer at a temperature of 25 °C. Minimum amount of dimethyl sulfoxide (1%) was employed to maintain monomeric form of **PTZ1** during experiment. The concentration of the salmon sperm DNA (Sigma, USA) stock solution was determined from the reported molar absorptivity at 260 nm (6600 M^−1^ cm^−1^). Spectroscopic studies were conducted by maintaining the concentrations of the compounds at a constant value (for UV-Vis spectroscopic studies: [MB] = 1.38 × 10^−5^ M, [**PTZ1**] = 6.9 × 10^−5^ M, [**PTZ2**] = 2.0 × 10^−5^ M; for fluorescence spectroscopic studies: [**PTZ1**] = 2.37 × 10^−5^ M, [**PTZ2**] = 1.18 × 10^−5^ M) while varying the concentration of DNA. 

Fluorescence quenching study by ferrocyanide ions were performed using 7.91 × 10^−5^ M stock buffer solution of K_4_(Fe(CN)_6_)) (final concentration in cuvette 7.91 × 10^−6^ M).

## 4. Conclusions

Two new 3,7-bis(aryl-amino)phenothiazine derivatives containing two primary amine or two sulfo groups were synthesized. It was shown by UV-Vis fluorescence spectroscopy that substituents in the arylamine fragment (located in the 3 and 7 positions of the phenothiazine fragment) play a crucial role in aggregation properties and interaction with DNA. Therefore, it is possible to control these properties by rational choice of the substituents in the arylamine fragment. For the **PTZ1** derivative containing amine fragments, the formation of H-aggregates with λ_max_ = 630 nm was demonstrated by UV-Visible spectroscopy, and confirmed by DFT calculations (BP86/def2-SVP, def2/J), i.e., the energy of the H-dimer was less than the energy of distant molecules. Interaction with DNA leads to dimer decomposition which results in hyperchromic effect and a bathochromic shift of 39 nm (λ_max_ = 669 nm). The interaction of monomeric forms of **PTZ1** with DNA (groove binding and intercalation) was studied. In the case of the **PTZ2** derivative containing sulfonic acid fragments, no aggregation and DNA binding is registered, which is explained by the electrostatic repulsion of sulfonic acid fragments. The obtained results open significant opportunities for the development of new drugs and photodynamic agents.

## Data Availability

The data presented in this study are available in Appendix A. Bindfit data are available at http://app.supramolecular.org/bindfit/view/Fit ID (accessed on 3 May 2021). Fit ID for MB: d69618fd-1929-4927-a7c2-db0d1acbe625, for **PTZ1**: da738fe0-c26d-4336-a3bd-5c7abcfcf7dd.

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
