# Peer review of "Arylamine Analogs of Methylene Blue: Substituent Effect on Aggregation Behavior and DNA Binding"

_ijms, 2021, doi:10.3390/ijms22115847_

Round 1
Reviewer 1 Report
In the manuscript, the authors prepared of bis(arylamino)phenothiazine derivatives and examined their aggregation and DNA-binding properties by using absorption and fluorescent spectra. The referee recommends it to be published in Int. J. Mol. Sci. after addressing the minor points listed below. In the fluorescence quenching at page 9, the authors should refer Figure 5A, B, C, and D in the main text. In Figure 5D, the fluorescence spectrum at the ratio of [PTZ1]/[DNA] = 0. 2 is shown with the red line, which seems to be weaker than that at the ratio of [PTZ1]/[DNA] = 0.1 of the green line in Figure 5B. The author should comment on the inconsistency. Proton NMR data of PTZ1 at page 11 should be carefully checked because of missing some protons. At page 4, the authors comment on the characteristic 1H NMR signals for PTZa and PTZ2, but not for PTZ1. In Figure S5 of SI, 13C NMR of PTZ1 should be replaced with good S/N one. The position number of the series of PTZ structures shown in SI (Figures S1-S3) should make correspond to that in Scheme 1.
Author Response
Dear Reviewer! Thank you very much for carefully reading and reviewing our paper! Unfortunately, our manuscript doesn’t contain above-mentioned paragraphs and table.
Reviewer 2 Report
In the manuscript, the authors prepared of bis(arylamino)phenothiazine derivatives and examined their aggregation and DNA-binding properties by using absorption and fluorescent spectra. The referee recommends it to be published in Int. J. Mol. Sci. after addressing the minor points listed below.
In the fluorescence quenching at page 9, the authors should refer Figure 5A, B, C, and D in the main text.
In Figure 5D, the fluorescence spectrum at the ratio of [PTZ1]/[DNA] = 0.2 is shown with the red line, which seems to be weaker than that at the ratio of [PTZ1]/[DNA] = 0.1 of the green line in Figure 5B. The author should comment on the inconsistency.
Proton NMR data of PTZ1 at page 11 should be carefully checked because of missing some protons. At page 4, the authors comment on the characteristic 1H NMR signals for PTZa and PTZ2, but not for PTZ1.
In Figure S5 of SI, 13C NMR of PTZ1 should be replaced with good S/N one.
The position number of the series of PTZ structures shown in SI (Figures S1-S3) should make correspond to that in Scheme 1.
Author Response
“In the manuscript, the authors prepared of bis(arylamino)phenothiazine derivatives and examined their aggregation and DNA-binding properties by using absorption and fluorescent spectra. The referee recommends it to be published in Int. J. Mol. Sci. after addressing the minor points listed below.
Answer:
Dear Reviewer! Thank you very much for carefully reading and reviewing our paper!
- In the fluorescence quenching at page 9, the authors should refer Figure 5A, B, C, and D in the main text.”
Answer: We have added references to Figure 5A, B, C and D in the main text (lines 264, 265, 279, 283).
“2. In Figure 5D, the fluorescence spectrum at the ratio of [PTZ1]/[DNA] = 0.2 is shown with the red line, which seems to be weaker than that at the ratio of [PTZ1]/[DNA] = 0.1 of the green line in Figure 5B. The author should comment on the inconsistency.”
Answer: Indeed, fluorescence spectra on Figure 5 were obtained at different instrument settings. We replaced them with spectra obtained at identical instrument settings and corrected slits width on line 313.
“3. Proton NMR data of PTZ1 at page 11 should be carefully checked because of missing some protons. At page 4, the authors comment on the characteristic 1H NMR signals for PTZa and PTZ2, but not for PTZ1.”
Answer: We checked Proton NMR data of PTZ1 and corrected misprints. We also add the sentence about characteristic 1H NMR signals for PTZ1 (line 116):
“In 1H NMR spectra of PTZ1 signals of the aromatic protons upfield shifted to 6.61-7.84 ppm due to electron donor effect of NH2 groups, caused by partial deprotonation with solvent. Signals of protons of NH groups are also shifted to 8.49 ppm.”
“4. In Figure S5 of SI, 13C NMR of PTZ1 should be replaced with good S/N one.”
Answer: We replaced Figure S5 of SI, 13C NMR of PTZ1 with better S/N one.
“5. The position number of the series of PTZ structures shown in SI (Figures S1-S3) should make correspond to that in Scheme 1.”
Answer: We suppose that referee meant Scheme 2, where synthetic route for the preparation of PTZa, PTZ1, PTZ2 was presented. We corrected Scheme 2 according to following order of the series of PTZ structures: (1) PTZa, (2) PTZ1, (3) PTZ2. The order of the series of PTZ structures shown in SI is the same.